# Classification of Alzheimer’s Disease Based on Weakly Supervised Learning and Attention Mechanism

**DOI:** 10.3390/brainsci12121601

**Published:** 2022-11-23

**Authors:** Xiaosheng Wu, Shuangshuang Gao, Junding Sun, Yudong Zhang, Shuihua Wang

**Affiliations:** 1School of Computer Science and Technology, Henan Polytechnic University, Jiaozuo 454000, China; 2School of Computing and Mathematical Sciences, University of Leicester, Leicester LE1 7RH, UK

**Keywords:** weakly supervised, attention module, classification, data augmentation

## Abstract

The brain lesions images of Alzheimer’s disease (AD) patients are slightly different from the Magnetic Resonance Imaging of normal people, and the classification effect of general image recognition technology is not ideal. Alzheimer’s datasets are small, making it difficult to train large-scale neural networks. In this paper, we propose a network model (WS-AMN) that fuses weak supervision and an attention mechanism. The weakly supervised data augmentation network is used as the basic model, the attention map generated by weakly supervised learning is used to guide the data augmentation, and an attention module with channel domain and spatial domain is embedded in the residual network to focus on the distinctive channels and spaces of images respectively. The location information enhances the corresponding features of related features and suppresses the influence of irrelevant features.The results show that the F1-score is 99.63%, the accuracy is 99.61%. Our model provides a high-performance solution for accurate classification of AD.

## 1. Introduction

As we all know, there is a kind of progressive neurodegenerative disease called Alzheimer’s disease. Among dementia patients over 60 years old, AD patients account for as high as 60–80% [1]. The vast majority of AD patients seek medical attention after various symptoms appear, but in general, the disease is discovered at an advanced stage. At present, there is no cure for AD, because we don’t know its etiology. In fact, brain characteristics associated with AD begin to change before the cognitive decline begin. Early and accurate diagnosis of AD is conducive to delaying the disease progression, but it requires us to find the disease as soon as possible and cooperate with drug treatment [2]. One of the three major dilemmas in Alzheimer’s diagnosis is the lack of early signal recognition. In clinical practice, professional doctors with rich clinical experience generally rely on comprehensive analysis and diagnosis of AD, which requires a lot of manpower and material resources, and there may be misdiagnosis. The medical field urgently needs professional and intelligent equipment to assist doctors in AD diagnosis.

Today, there are many methods based on artificial intelligence and computer vision in neuroimaging, mainly including deep learning (DL) [3] methods and machine learning [4] methods. Traditional machine learning methods not only require manual feature extraction, but also have limitations. In order to eliminate the difficulties of traditional machine learning methods in the field of medical images, automatic feature extraction of DL is becoming more and more popular. In the detection of AD, the deep learning methods include unsupervised learning [5], supervised learning [6], and weakly supervised learning (WSL) [7].

The training data of unsupervised learning is unlabeled. Researchers mine information from a great quantity of unlabeled data. Autoencoders (AE) [8] and Restricted Boltzmann Machines (RBM) [9] are widely used in different applications of unsupervised feature representation learning. Li et al. [10] stacked multiple automatic encoders and proposed an automatic and effective stacked detection model to help doctors diagnose diseases. Lu et al. [11] expanded and improved the feature extraction ability of RBM through the introduction of redundancy removal mechanism, and realized fast and accurate classification. People use supervised learning more widely than unsupervised learning. CNN is a typical network in supervised methods, and also the most successful depth model in image analysis. There are many powerful CNN models for AD classification, such as AlexNet, ResNet, VGGNet, DenseNet, and Inception, etc. [12,13,14,15]. Zhang et al. [16] expanded 2D convolution to 3D to obtain spatial information, and extracted multi-scale features of data through dense connection, with an accuracy rate of 97.35%. Liu et al. [17] proposed a multitask model architecture based on CNN, and combined with patch features extracted from 3D DenseNet model to classify AD. Folego et al. [18] proposed a classification method named ADNet based on VGG, which is implemented end-to-end and realizes an automatic and fast learning process. Although CNN have outperformed most traditional feature extraction methods and provided SOTA solutions for different classification tasks, the supervised learning requires massive, high-quality manually annotated image data to accurately represent features, while image samples in the medical field are lacking is a persistent problem. Therefore, WSL method [19] is of great significance in low-cost and high-precision medical image data mining.

WSL processes medical images with limited labels to classify diseases. Lian et al. [20] designed an attention module based on weak supervision to automatically identify discriminant regions according to structural changes in the brain. Hu et al. [21] proposed a WSL framework, which can quickly implement multimodal image registration tasks using only input image pairs. Shuang et al. [22] proposed a deep network framework based on WSL. The framework consists of two networks namely, the backbone network with attention mechanism and the parallel task network for image classification and image reconstruction in parallel, which can recognize AD with limited annotation. However, the method still has difficulties in accurate feature selection. Compared with natural images, the adaptive ability of medical images is worse, because it contains more feature information. Therefore, when building the DL model, we should fully consider the features of images [23], build the most appropriate model, effectively locate and extract feature information.

In this study, our main contributions are as follows:We propose a new model that fuses weak supervision learning and an attention mechanism (WS-AMN) for AD classification.An attention module is introduced into ResNet50 residual block, which makes network focus on feature information and improves the discriminative ability of the backbone network.In AD classification results. F1-score is 99.63%, accuracy is as high as 99.61%. It shows excellent performance on OASIS dataset, which is important for AD classification.

The structure of this paper is as follows. The related works are described in Section 2. Materials and Method are introduced in the Section 3. Experiments is described in Section 4. In Section 5, we discuss the effectiveness of our approach. Finally, the conclusion is given in Section 6.

## 2. Related Works

### 2.1. Feature Extraction Network

Feature extraction is a key point of DL. Compared with traditional methods, DL can automatically extract features for AD classification. Nowadays, many classical feature extraction networks that can effectively classify AD, such as VGGNet, AlexNet, DenseNet, and Inception, which greatly improve the performance of classification tasks. These feature extraction networks combine feature extraction and classification. Different networks have their own unique characteristics. However, a large number of parameters and calculations will increase the training time of these models.

The residual network proposed by He et al. [24] uses residual learning to solve the degradation problem of deep networks so that we can train deeper networks. ResNet training has less computation and higher performance than VGGNet, AlexNet and GoogLeNet. Therefore, in this paper, we choose ResNet50 as basic feature extraction network and make further improvement on this basis.

### 2.2. Transfer Learning

At present, artificial intelligence technology requires massive data to support. Many studies train Deep Neural Network (DNN) [25] from scratch, which not only requires a lot of training time, but also has limited available resources. Especially in the medical field, because of its particularity, we can’t get as much data as natural images. Labeling of medical images is also a difficult task.

In most cases, transfer learning (TL) [26] is performed based on the existing model. The principle of TL is to train the network on large dataset, and then use the trained weight as the initial weight in the new target task to train the new model. TL can reduce the dependence on data and improve the generalization ability of models. The learning process is shown in Figure 1. We use ResNet50 pretrained model parameters as the initialization parameters of WS-AMN to help us reduce training time.

### 2.3. Weakly Supervised Learning

Deep learning models can help us achieve significant classification results without manual feature extraction. Supervised learning models rely on a large amount of labeled training data to obtain accurate classification, but the data labeling process is often lengthy, expensive, and error prone. Unsupervised learning does not use training data with labeled information. It directly finds intrinsic connection in a large amount of unlabeled data through network training, but it lacks target labels as guidance, the accuracy obtained by unsupervised learning is not ideal. Therefore, relevant researchers proposed the concept of WSL. Due to the high requirements of medical images for data labels, it’s difficulty to collect a large number of effective labels. WSL can not only reduce the workload of manual labeling, but also introduce proposed supervised information, which can make more efficient use of data, reduce the amount of labeling, and improve classification performance. Therefore, our study used WSL to classify AD.

### 2.4. Attention Module

With the research development of DL and the attention mechanisms of the visual system, more and more researchers tend to add attention mechanisms to the DL model to optimize the overall performance of networks. Spatial Transformer Networks (STNs) [27] can be trained end-to-end without changing the original network, thereby improving the robustness and accuracy of the network. SE-Net [28] proposes a squeeze-excitation attention mechanism, which uses 1 × 1 convolution for the channel domain in the side branch network layers to adjust weight parameters in main branch network layers.

In this paper, we use the Convolutional Block Attention Module (CBAM) [29], a further extension of Squeeze-and-Excitation module in SE-Net, to perform channel and spatial domain attention, respectively. CBAM combines Channel Attention Module (CAM) and Spatial Attention Module (SAM) to realize the fine distribution and processing of information. The relationship between CBAM and CAM, SAM is shown in Figure 2.

CAM: The input feature map F(H×W×C) performs global max pooling and global average pooling according to width and height respectively, the generated feature vectors are $F_{avg}^{C}$ and $F_{max}^{C}$ respectively. Then the two feature vectors are respectively entered into MLP, the element-wise sum operation is performed on their outputs, the channel attention feature map is generated by sigmoid activation, namely $M_{C}$.
(1)$$\begin{array}{cc} M_{c}\left(F\right) & =\sigma (MLP(AvgPool(F))+MLP(MaxPool(F)))\\ \; & =\sigma (W_{1}\left(W_{0}\left(F_{avg}^{c}\right)\right)+W_{1}\left(W_{0}\left(F_{max}^{c}\right)\right)) \end{array}$$

σ is the sigmoid activation function, the size of the hidden layer activation function in MLP is $R^{C/r\times 1\times 1}$, $W_{0}\in R^{C/r\times C}$ and $W_{1}\in R^{c/r\times C}$ are weights in MLP.

SAM: Multiply channel attention and input feature map *F* to get channel-refined feature map $F^{\prime }$. The feature map $F^{\prime }$ is the input feature map of the spatial module. Average pooling and max pooling are used to map RGB channels of the feature map, two feature maps of H×W×1 are generated. The two feature maps are concatenated according to the channel, the dimension is reduced by the convolution operation, that is H×W×1. The spatial attention feature $M_{s}$ is generated through sigmoid. Finally, multiply $M_{S}$ with $F^{\prime }$ to get the final feature.
(2)$$\begin{array}{cc} M_{s}\left(F^{\prime }\right) & =\sigma (f^{7\times 7}\left(\left[AvgPool\left(F^{\prime }\right);MaxPool\left(F^{\prime }\right)\right]\right)\\ \; & =\sigma (f^{7\times 7}\left(\left[{F^{\prime }}_{avg}^{s};{F^{\prime }}_{max}^{s}\right]\right)) \end{array}$$

σ is the sigmoid activation function, $f^{7\times 7}$ indicates the convolution operation with the convolution kernel of 7×7.

The channel-refined feature map $F^{\prime }$ and final feature map $F^{\prime \prime }$ are obtained as:(3)$$\left\{\begin{array}{c} F^{\prime }=M_{C}\left(F\right)⊗F\\ F^{\prime \prime }=M_{S}\left(F^{\prime }\right)⊗F^{\prime } \end{array}\right.$$

## 3. Materials and Method


### 3.1. Dataset

The AD dataset used in our work come from the OASIS (Open Access Series of Imaging Studies). The MR brain image samples of NC and AD in the dataset are shown in Figure 3. The dataset consisted of 416 subjects. We randomly selected 80 AD and 80 NC subjects, a total of 160 subjects. The grouping of AD and NC is determined by the Clinical Dementia Rating (CDR), 0 indicates NC, greater than 0 indicates AD. The scoring criteria is shown in Table 1. There are a lot of images for us to choose from in 3D MRI scanning, but selecting the best training data is still critical to the success of the method. Therefore, we calculate the image entropy of each slice and select the slice with a large amount of information as the training data. Image with probabilities $p_{1},p_{2},\cdots ,p_{M}$, the image entropy is calculated as shown in formula (Equation 4).
(4)$$H=-\sum_{g=1}^{M}p_{g}logp_{g}$$

Entropy provides a measure of change in each slice. The pictures are sorted in descending order according to entropy. The image with the highest entropy value can be regarded as the image with the most information. We use the entropy-based ranking mechanism to pick the 32 most informative images from the axial plane of each 3D scan.

In Table 2, we divide the dataset in detail. Each category uses 2560 images. We divide the dataset into 8:1:1, that is, 2048 training sets, 256 validation sets and 256 test sets for each category.

### 3.2. WS-AMN

The training process of WS-AMN is shown in Figure 4. WS-AMN combines WSL and an attention mechanism. The input images first generate feature maps and attention maps through the feature extraction network of line ➀. The feature extraction network is a residual network that introduces CAM and SAM. Then the network randomly selects an attention map, and the attention map guides the data for attention cropping and attention dropping, corresponding to line ➁. The augmented images and the original images are jointly input into the network for training. The feature maps and the attention maps obtain the feature matrix through a bilinear attention pooling algorithm, corresponding to line ➂. The feature matrix will be used as the input of linear classification layer. Finally, classification layer outputs classification results of AD or NC.

In the testing phase, the total classification probability is composed of two parts: the coarse classification probability and the fine classification probability. The original images obtain attention maps and feature maps through the feature extraction network. Then the feature matrix is obtained by multiplying the feature map and attention map points with BAP algorithm. The feature matrix for pooling and classification operations. Finally, we obtain the coarse classification probability, corresponding to line ➀ in Figure 5. The fine classification probability is that after the network obtains the attention maps, the attention maps are added to obtain the sum of the attention, the target region is cropped in combination with the feature maps, the cropped result is also sent to the test network. Finally, feature maps and attention maps perform element-wise multiplying, pooling and classification operations to get the fine classification probability. The acquisition of the fine classification probability corresponds to line ➁ in Figure 5. The final probability value P is the average of the coarse classification probability $P_{c}$ and the fine classification probability $P_{f}$.

### 3.3. Bilinear Attention Pooling

Bilinear pooling is mainly used for feature fusion. The bilinear pooling proposed by Lin et al. [30] is aimed at two different features extracted from the same sample, then the two feature vectors are fused by bilinear pooling. In WSDAN [31], the feature fusion of the feature maps and the attention maps is performed through bilinear attention pooling (BAP). BAP combines the attention maps and the feature maps, which makes it easier to increase the number of attention regions and thus improve classification accuracy. The process of BAP is shown in Figure 6.

First, feature maps $F\in R^{H\times W\times M}$ and attention maps $A\in R^{H\times W\times N}$ generated by the backbone network are fused, *W*, *H* and *M* are the width, height and number of channels of the feature layers, respectively. The attention maps are obtained by formula (Equation 5).
(5)$$A=f\left(F\right)=\overset{N}{\underset{i=1}{⋃}}A_{i}$$
*f*(·) is the convolutional function, *N* represents the number of attention maps, and $A_{i}\in R^{H\times W}$ represents the *i*-th attention map.

Then we use the attention maps to guide the redistribution of each element of the feature maps, and use the BAP algorithm to perform feature fusion, feature maps and attention maps are multiplied element-wise to generate partial feature maps $F_{i}$. As shown in formula (Equation 6).
(6)$$F_{i}=A_{i}⨀F\left(i=1,2,\ldots ,N\right)$$

⨀ means element-wise multiplication.

In order to avoid the excessively high dimension of $F_{i}$ after feature fusion, the feature extraction function *g*(·) is used to reduce the dimension. The global average pooling of each group of eigenvalue maps is performed to further extract part of the features. We obtain the *i*-th attention feature $f_{i}\in R^{1\times N}$.
(7)$$f_{i}=g\left(F_{i}\right)$$

Finally, $F_{i}$ is summed to obtain the feature matrix $P\in R^{N\times M}$, as shown in formula (Equation 8). The resulting feature matrix *P* is also the input for linear classification.
(8)$$P=\Gamma \left(A,F\right)=\left(\begin{array}{c} g(a_{1}⨀F)\\ g(a_{2}⨀F)\\ \cdots \\ g(a_{i}⨀F)\\ \cdots \\ g(a_{N}⨀F) \end{array}\right)=\left(\begin{array}{c} f_{1}\\ f_{2}\\ \cdots \\ f_{i}\\ \cdots \\ f_{N} \end{array}\right)$$

Γ(A,F) represents BAP between attention maps and feature maps. i∈[0,1]. $a_{i}$ represents a characteristic part of an attention map.

### 3.4. Attention-Guided Data Augmentation

Each original image enters the feature extraction network to generate attention maps, we randomly select an attention map $A_{i}$ to guide the data augmentation process [31], and normalize $A_{i}$ as *i*-th augmentation map $A_{i}^{*}\in R^{H\times W}$. $A_{i}$ is normalized to eliminate the interference of singular samples on the whole data. Then the methods of attention cropping and attention dropping are used for effective data augmentation.
(9)$$A_{i}^{*}=\frac{A_{i}-\min A_{i}}{\max A_{i}-\min A_{i}}$$
min⁡ $A_{i}$ is the smallest pixel value of the i−th attention map, max⁡ $A_{i}$ is the largest pixel value of the i−th attention map.

#### 3.4.1. Attention Cropping

Attention cropping is to crop the feature regions concerned by the network, and then we enlarge the regions to extract more detailed local features. $C_{i}\left(m,n\right)=1$, when element $A_{i}^{*}\left(m,n\right)$ is greater than the threshold $\theta_{c}\in \left[0,1\right]$, otherwise, $C_{i}\left(m,n\right)=0$. We find a smallest bounding box $B_{i}$ that can cover the region of crop mask $C_{i}$, and enlarge the region from original image as the input data. The attention cropping process is shown in formula (Equation 10).
(10)$$C_{i}\left(m,n\right)=\left\{\begin{array}{cc} 1, & if\;A_{i}^{*}\left(m,n\right)>\theta_{c}\\ 0, & otherwise \end{array}\right.$$

#### 3.4.2. Attention Dropping

Attention dropping is to drop the current feature regions of network attention, so that the regions are no longer attended. Attention dropping can improve the probability of identifying other key attention regions, and prevent multiple attention maps from focusing on the same feature region. We set the element $A_{i}^{*}\left(m,n\right)$ larger than the threshold $\theta_{d}\in \left[0,1\right]$ to 0 and the others to 1, we obtain the drop mask $D_{i}$. The attention dropping process is shown in formula (Equation 11).
(11)$$D_{i}\left(m,n\right)=\left\{\begin{array}{cc} 0, & if\;A_{i}^{*}\left(m,n\right)>\theta_{d}\\ 1, & otherwise \end{array}\right.$$

### 3.5. Attention Regularization

In order to make the features of the same part as similar as possible, we use attention regularization loss [31] to supervise the learning process of attention. The attention regularization loss is designed so that each feature map is fixed at the center of each part, and part features $f_{i}$ will be close to the global feature center $c_{i}$. so that there will be no large difference between the generated attention maps. The attention map $A_{i}$ will be activated on the same *i*-th object part. The loss function $L_{A}$ is defined as shown in formula (Equation 12).
(12)$$L_{A}=\sum_{i=1}^{N}{∥f_{i}-c_{i}∥}_{2}^{2}$$
where $c_{i}$ represents the *i*-th feature center. $c_{i}$ is initialized to 0 and updated during model training according to the sliding average formula (Equation 13).
(13)$$c_{i}\leftarrow c_{i}+\beta \left(f_{i}-c_{i}\right)$$
where β is used to control the update rate of $c_{i}$.

### 3.6. Improved ResNet50 Model

Deep residual networks have been widely used in various feature extraction applications. The deeper the network layers, the deeper features can be obtained, and the stronger the expression ability. From the 5 layers of LeNet at the beginning to the 8 layers of AlexNet, and then to the 19 layers of VGG19, GoogleNet has a total of 22 layers. However, when the network reaches a certain depth, with the deepening of the number of CNN layers, the classification performance will not continue to improve. Problems such as random gradient explosion and gradient disappearance will also reduce network accuracy, residual network is proposed to solve those problems. In the residual networks, even if the number of network layers increases, the features expressed will be better, and the performance will be stronger. In the residual blocks, 1×1 convolution is used to reduce the amount of computation.

ResNet50 is used as feature extraction network. Small differences between MRI of AD and NC. In order to extract more feature information, we improved ResNet50, as shown in Figure 7. The attention mechanism is introduced into residual blocks. The improved residual block enables the network to obtain more details related to the target, while ignoring other irrelevant information, and enhance feature extraction ability.

The output feature matrix obtained by the improved residual block is $\bar{X}$, *F* represents the residual maps after three layers of convolution, $F^{\prime \prime }$ is the final feature maps after the attention module.
(14)$$F^{\prime }=M_{C}\left(F\right)⊗F$$
(15)$$F^{\prime \prime }=M_{S}\left(F^{\prime }\right)⊗F^{\prime }$$
(16)$$\bar{X}=X+F^{\prime \prime }$$

### 3.7. Evaluation Metrics

In order to evaluate the performance of the model, we choose five common metrics of sensitivity, specificity, precision, accuracy, and F1-score as the performance test metrics of WS-AMN. We use the visualization tool, confusion matrix, to obtain these metrics. The positive category is AD, and the negative category is NC.

Sensitivity. Results of correct recognition of AD by the model.
(17)$$\mathrm{Sensitivity}=\frac{\mathrm{True}\;\mathrm{Positive}}{\mathrm{True}\;\mathrm{Positive}+\mathrm{False}\;\mathrm{Negative}}$$

Specificity. The proportion of negative samples judged as true negatives by the model. The greater the specificity, the fewer results of normal people diagnosed with AD.
(18)$$\mathrm{Specificity}=\frac{\mathrm{True}\;\mathrm{Negative}}{\mathrm{False}\;\mathrm{Positive}+\mathrm{True}\;\mathrm{Negative}}$$

Precision. The proportion of true positives among all the results identified as positive.
(19)$$\mathrm{Precision}=\frac{\mathrm{True}\;\mathrm{Positive}}{\mathrm{True}\;\mathrm{Positive}+\mathrm{False}\;\mathrm{Positive}}$$

Accuracy. Reflects the proportion of correctly classified AD and NC.
(20)$$\mathrm{Accuracy}=\frac{\mathrm{True}\;\mathrm{Positive}+\mathrm{True}\;\mathrm{Negative}}{\mathrm{True}\;\mathrm{Positive}+\mathrm{False}\;\mathrm{Negative}+\mathrm{True}\;\mathrm{Negative}+\mathrm{False}\;\mathrm{Positive}}$$

F1-score. It is a harmonic mean of precision and sensitivity. A high F1-score can be obtained when both the precision and sensitivity are high.
(21)$$F1-\mathrm{score}=\frac{2\;\mathrm{True}\;\mathrm{Positive}}{2\;\mathrm{True}\;\mathrm{Positive}+\mathrm{False}\;\mathrm{Positive}+\mathrm{False}\;\mathrm{Negative}}$$

## 4. Experiments

### 4.1. Obtaining Attention Maps via Weakly Supervised Learning

First, the attention maps are obtained through a weakly supervised learning network, and then the attention maps are used to guide data augmentation. The visualization results of data augmentation are shown in Figure 8. The visualization results show that attention cropping can effectively crop the feature regions without excessive background regions. Attention dropping erases the current feature regions and ensures that other regions can receive more attention. Effective data augmentation can enhance the robustness of the network.

### 4.2. Selection of Backbone Network

First, we verified the effectiveness of the selected feature extraction network, we compare the classical CNNs is shown in Table 3.

We use different CNNs to classify AD. Under the same conditions, the accuracy of ResNet50 is the highest 92.58%. Compared with VGG19, VGG19 and Inception_V3, ResNet50 is 6.01%, 5.86% and 3.32% higher, respectively, significantly better than the two popular classification networks. The accuracy of ResNet50 and ResNet101are similar, but the volume of ResNet50 is smaller and the amount of calculation is less.

### 4.3. Evaluation of Different Feature Extraction Networks

We use WSDAN model and perform weakly supervised guided data augmentation, the enhanced images are sent to the network together with the original images, and three different models are used for training. The results of the five metrics are listed in Table 4.

It is not difficult to see that after weakly supervised data augmentation, the three feature extraction methods have very obvious improvements. The accuracy of VGG16 is improved by 10.19%, VGG19 is improved by 10.35%, Inception_V3 is improved by 9.81%, ResNet50 is improved by 6.44%, and ResNet101 is improved by 6.66%. In the basic model of WSDAN, ResNet50 achieves the best performance overall, precision is 1.75%, 3.72% higher than Inception_V3, VGG19. The F1-sorce, ResNet50 is also 2.03% higher than VGG16. We can see that ResNet50 can extract features more fully. Therefore, ResNet50 network is selected for feature extraction.

### 4.4. Evaluation of WS-AMN

In Table 5. We use improved ResNet50 feature extraction network for training WS-AMN.

It can be seen from the experimental results that the five metrics have been improved compared with WSDAN, and the accuracy is as high as 99.61%, which indicates that WS-AMN model can almost correctly classify AD and NC. It shows that this method has good classification accuracy. Figure 9 shows the visualization results of WSDAN and WS-AMN. We can see that the feature extraction range of the improved network is larger, more refined, and contains more effective information.

### 4.5. Comparison with State-of-the-Art Approaches

We compared WS-AMN with six SOTA methods (ADVIAN, 3D-EB+SVM-Pol, PZM+LRC, CNN-RNN-LATM, GLCM-ELM, 5L-CNN). Data of all methods are based on slices. The sensitivity, specificity, precision, accuracy and F1-score results of each model are shown in Table 6.

From the overall results of the five indicators in Figure 10, WS-AMN achieved the best performance. Accuracy is 99.61%, followed by ADVIAN (97.76%), 5L-CNN (94.39%), PZM + LRC (93.06%), CNN-RNN-LSTM (92.50%), GLCM-ELM (92.30%), 3D-EB + SVM-Pol (92.04%). Other metrics are also higher than the six SOTA methods, which can see the effectiveness of our method.

## 5. Discussion

The early and accurate diagnosis of Alzheimer’s disease is very important for disease prevention. In this paper, we propose WS-AMN that fuses weak supervision and attention mechanism. CNN is the most successful deep model in image analysis, which provides help for image-aided classification. As the depth of CNN continues to deepen, a series of problems follow, such as gradient disappearance, gradient explosion and so on. The emergence of ResNet alleviates these problems, the residual block solves the problem of network degradation, and the skip connection makes the network deeper, showing excellent performance. In this article, we compare the classification accuracy of different CNN networks. Compared with VGG16, VGG19, Inception_V3 and ResNet101, ResNet50 has the highest accuracy rate of 92.58%. We introduce attention mechanism into the residual block of the backbone network, so that the network not only focuses on the key feature areas to improve the classification accuracy, but also generates an attention map to guide attention cropping and attention dropping. Enter the network together for training to enhance the data. Then the feature map generated by the backbone network and the attention map are fused through BAP. It is obvious from the heat map in Figure 9 that our WS-AMN is able to extract more and more accurate effective features.

To further evaluate the effectiveness of WS-AMN, we compare with six SOTA methods. From the results of five indicators, the WS-AMN obtained the best result of sensitivity of 99.61%, specificity of 99.85%, precision of 99.63%, accuracy of 99.61% and F1-score of 99.63%. Compared with the random data augmentation used by ADVIAN and 5L-CNN, our weakly supervised guided data augmentation indeed achieves better results. Therefore, our proposed WS-AMN method is crucial for the accurate diagnosis of AD.

Compared with previous methods [19,23,38] of data augmentation and attention to extract key features, our WS-AMN is a more effective method, which can automatically focus on key feature regions for purposeful data augmentation. The experimental part demonstrates the effectiveness of our method from several aspects. Although our method achieves high accuracy in AD classification, its performance and real-world use can be further improved in the future. We only used the cross-sectional in OASIS, which is relatively single, and the two types of data are very average, without considering the impact of data differences, so we will conduct experiments with multiple data sets in the future. Our network currently only focuses on the classification of AD and NC. In the future, mild cognitive impairment (MCI) should also be taken into account. MCI is an intermediate stage between NC and AD, with a high probability of turning to AD. Therefore, accurate detection of MCI can further effectively prevent AD. Our network can now achieve a classification accuracy of 99.61% for AD. In the future, pipelines should be developed for average users. By monitoring brain changes, we can predict whether they will develop AD in the upcoming 10 years.

## 6. Conclusions

We propose a DL model WS-AMN that fuses an attention mechanism and WSL for AD classification. The proposed WS-AMN achieves excellent performance in the five metrics of sensitivity, specificity, precision, accuracy, and F1-sorce, the accuracy is as high as 99.61%. The excellent performance shows that the proposed WS-AMN model is suitable for AD classification. The model with attention is better than that without attention mechanism in the classification of Alzheimer’s disease. In the medical field, acquiring large amounts of data and high-quality medical image annotation is time-consuming and expensive. We use WSL method to solve the problem of small AD dataset, and combine the attention mechanism to effectively perform feature mining. Our method provides a new idea for AD classification. This study effectively improves the accuracy of AD automatic classification, which is of great significance to accurately identify and diagnose diseases in the medical field.

## Figures and Tables

**Figure 1 brainsci-12-01601-f001:**
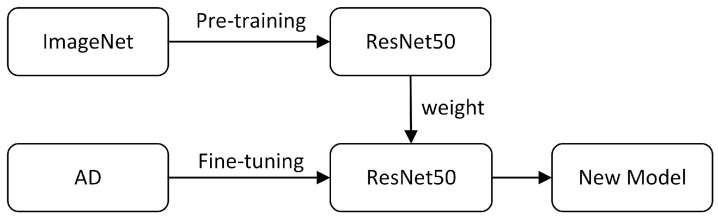
Transfer learning.

**Figure 2 brainsci-12-01601-f002:**
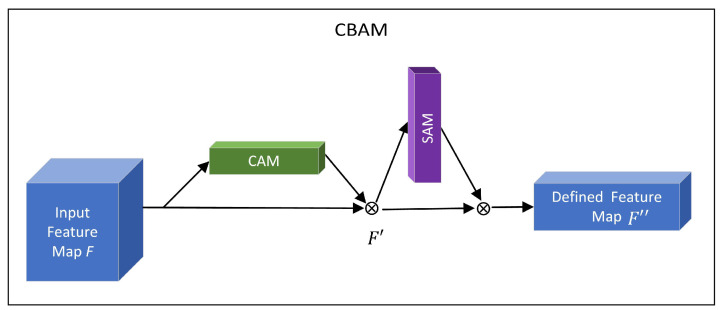
Relationship between CBAM and CAM, SAM module.

**Figure 3 brainsci-12-01601-f003:**
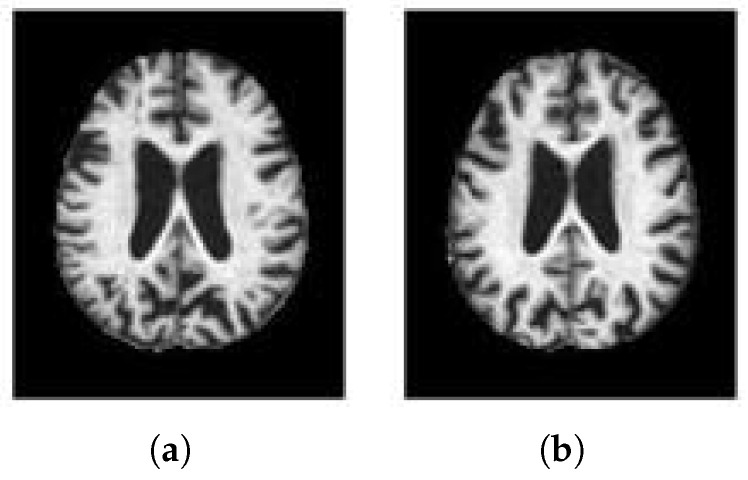
Samples of OASIS dataset. (**a**) AD. (**b**) NC.

**Figure 4 brainsci-12-01601-f004:**
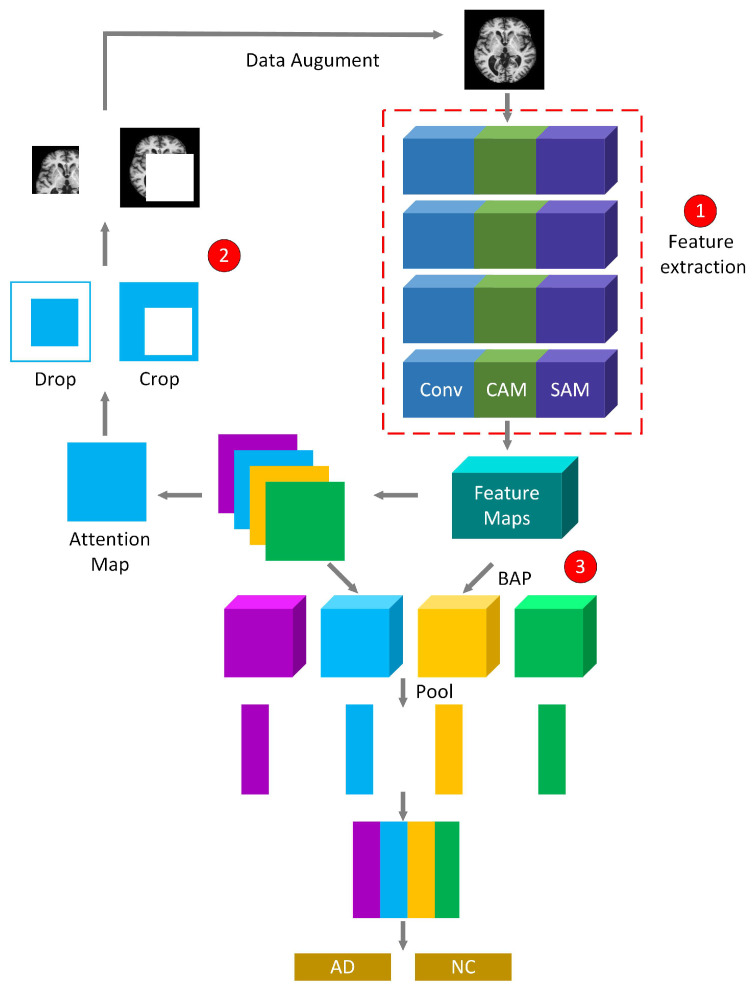
The training process of WS-AMN.

**Figure 5 brainsci-12-01601-f005:**
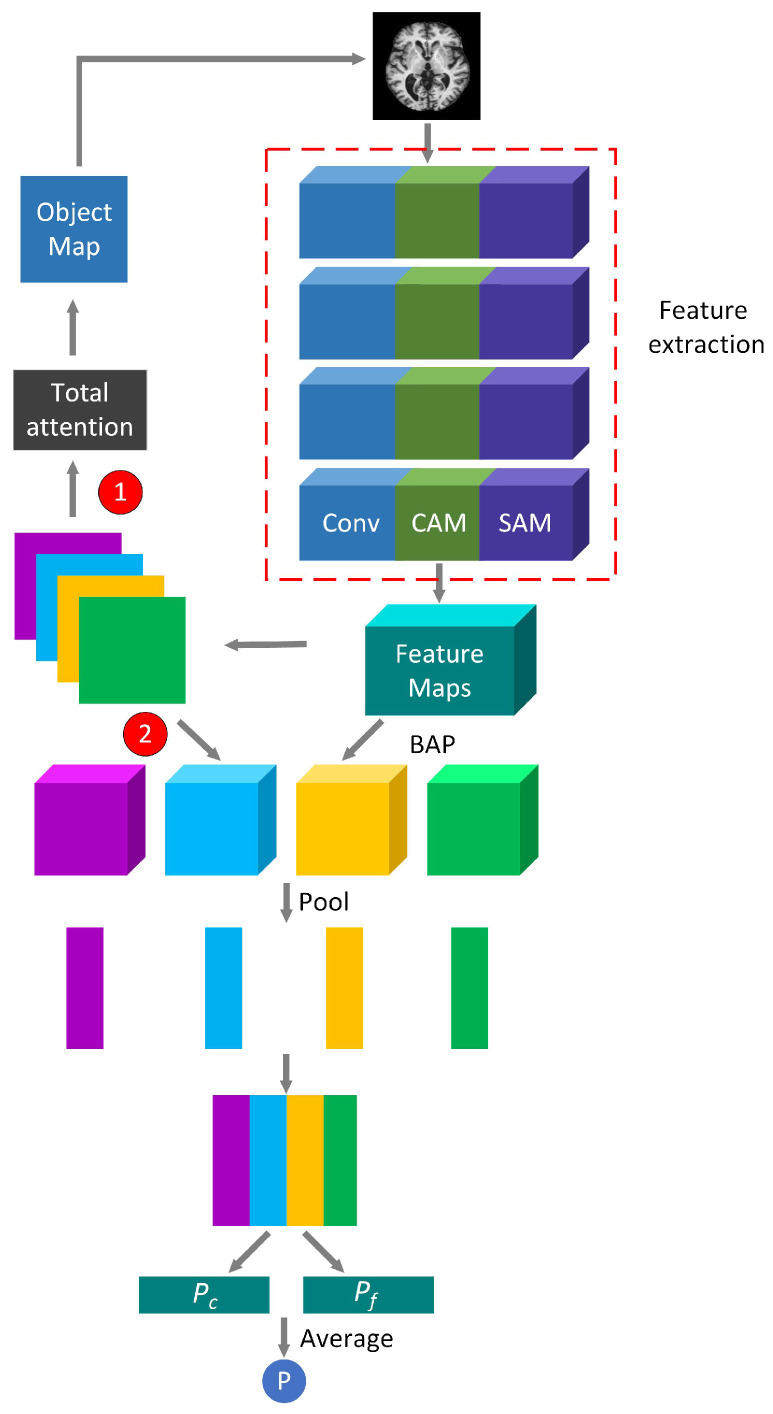
The test process of WS-AMN.

**Figure 6 brainsci-12-01601-f006:**
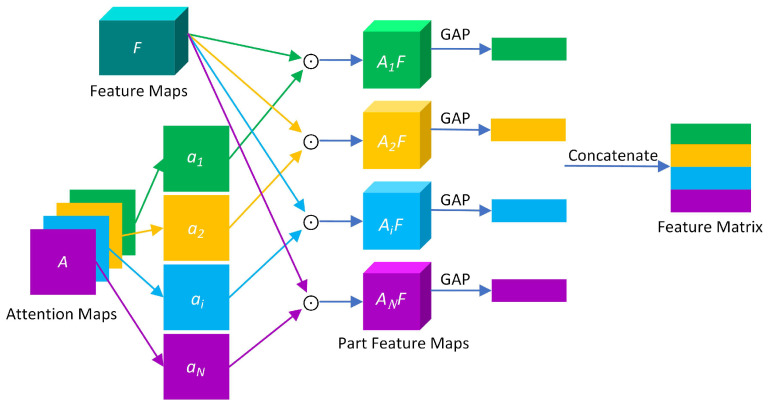
The process of BAP.

**Figure 7 brainsci-12-01601-f007:**
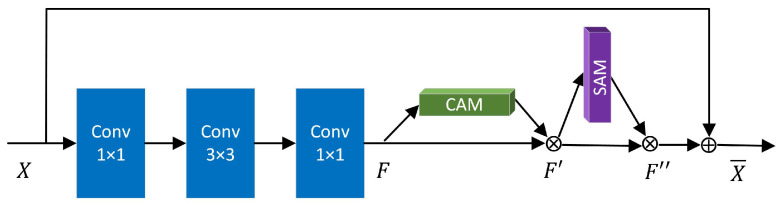
Improved residual block.

**Figure 8 brainsci-12-01601-f008:**
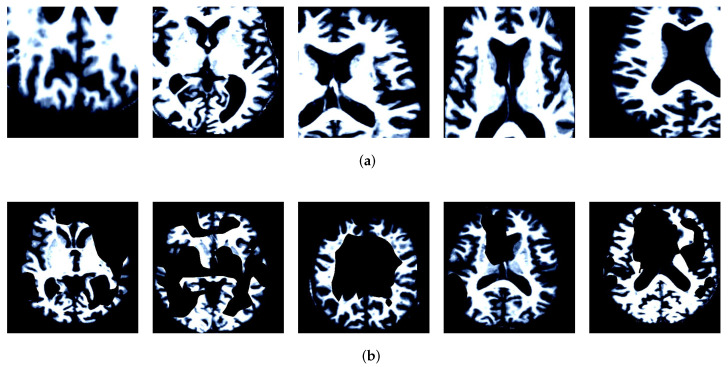
Visualization results of data augmentation (**a**) Attention Cropping. (**b**) Attention Dropping.

**Figure 9 brainsci-12-01601-f009:**
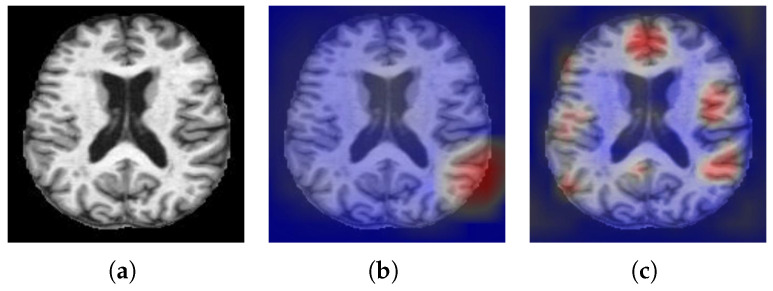
Visual comparison (**a**) Input image. (**b**) WSDAN. (**c**) WS-AMN.

**Figure 10 brainsci-12-01601-f010:**
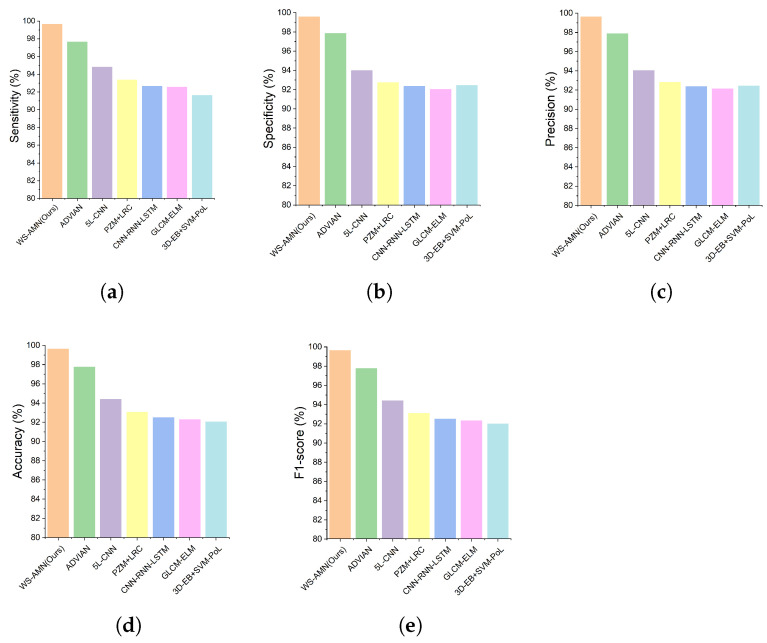
Comparison of the (**a**) sensitivity, (**b**) specificity, (**c**) precision, (**d**) accuracy and (**e**) F1-score.

**Table 1 brainsci-12-01601-t001:** Diagnostic criteria for CDR.

CDR 0	CDR 0.5	CDR 1.0	CDR 2.0	CDR 3.0
Nondemented	Suspected dementia	Mild dementia	Moderate dementia	Severe dementia

**Table 2 brainsci-12-01601-t002:** Division of dataset.

	NC	AD	Total
Training	2048	2048	4096
Validation	256	256	512
Testing	256	256	512
Total	2560	2560	5120

**Table 3 brainsci-12-01601-t003:** Traditional classification networks.

Model	Accuracy (%)
VGG16	86.57
VGG19	86.72
Inception_V3	89.26
ResNet50	92.58
ResNet101	92.16

**Table 4 brainsci-12-01601-t004:** WSDAN basic network.

Feature Extraction Network	Sensitivity (%)	Specificity (%)	Precision (%)	Accuracy (%)	F1-Score (%)
VGG16	96.25	98.16	97.88	96.76	97.05
VGG19	98.75	95.59	95.18	97.07	96.93
Inception_V3	99.58	97.43	97.15	98.44	98.35
ResNet50	99.26	98.75	98.90	99.02	99.08
ResNet101	99.58	98.52	98.35	98.82	98.76

**Table 5 brainsci-12-01601-t005:** Comparison of WSDAN and WS-AMN.

Model	Sensitivity (%)	Specificity (%)	Precision (%)	Accuracy (%)	F1-Score (%)
WSDAN	99.26	98.75	98.90	99.02	99.08
WS-AMN (Ours)	99.63	99.58	99.63	99.61	99.63

**Table 6 brainsci-12-01601-t006:** Comparison with SOTA approaches.

Approach	Sensitivity (%)	Specificity (%)	Precision (%)	Accuracy (%)	F1-Score (%)
ADVIAN [32]	97.65	97.86	97.87	97.76	97.75
3D-EB + SVM-Pol [33]	91.63	92.45	92.42	92.04	92.00
PZM + LRC [34]	93.37	92.76	92.83	93.06	93.08
CNN-RNN-LSTM [35]	92.65	92.35	92.38	92.50	92.51
GLCM-ELM [36]	92.55	92.04	92.13	92.30	92.31
5L-CNN [37]	94.80	93.98	94.04	94.39	94.41
WS-AMN (Ours)	99.63	99.58	99.63	99.61	99.63

## Data Availability

We using open datasets to tested our method. The OASIS dataset can be found in http://www.oasis-brains.org/ (accessed on 5 September 2021).

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
