# Peer review of "Classification of Alzheimer’s Disease Based on Weakly Supervised Learning and Attention Mechanism"

_brainsci, 2022, doi:10.3390/brainsci12121601_

Round 1

Reviewer 1 Report

General comment:

This manuscript puts forth a methodology using deep learning to classify Alzheimer’s disease using images from 3D fMRI scans. The authors propose a weakly supervised learning model used to augment data fused with an attention module with various domains separating information from various locations of the image. This leads to very impressive results with exceptionally high >99% in F1-scoring and accuracy levels. The data they used seems to be thoroughly explained and they go into detail on the architecture, even though improvements are always welcome. The conclusions are clear and they provide comparisons with other methods, showing that their accuracy is significantly higher.

The authors put in a lot of work; the approach fuses various technical deep-learning methodologies for which the applicability is obvious. However, plenty of work has already been done using deep learning in the classification of Alzheimer's disease and even if more tools should provide benefits and potentially added value, from a research perspective, it would be worthwhile to better understand WHY this current approach proposed by the authors is novel or how does it provide added value for people wanting to classify AD.

In my opinion, there are several ways in which the paper can be improved and I explain below, if the authors want to consider them.

Comments:

1. An extremely similar approach and methodology have been employed by Liang and Hu (2020) – whom the authors cited. Please explain and provide arguments as to why this methodology is a) novel as compared to the ones from Liang and Hu (2020) and b) why does it bring added value of applicable nature?

Liang, S.; Gu, Y. Computer-aided diagnosis of Alzheimer’s disease through weak supervision deep learning framework with 450 attention mechanism. Sensors 2020, 21, 220

2 Would it be easier for anyone to use this methodology as opposed to the previous work done? Could an average person monitor changes in brain structure (if they schedule fMRI scans for example) and use the tool you propose to “predict” with >99% accuracy if they will develop AD in the upcoming 10 years?

As the authors mention in the Introduction, early warning detection of the disease is absolutely vital for AD. Novelty and added value can be argued through applicability for example. Have the authors explored the opportunity of developing a pipeline for the “average user”? I’m not saying this should be done now, but they can set forth the pillars for this type of pipeline in the future.

3. Please provide a more thorough explanation of how the network architecture is implemented.

a. For example, sub-section 4.1 could be better explained by placing numbers in Figure 4 that explain chronologically how the process takes place, as it is not quite clear. It starts from the upper right section – what are those 4 (blue, green, purple) layers that lead to Conv, CAM, and BAM? You only described CBAM – is that BAM? What happens in the rest? It seems straightforward, but details are always welcome for reproducibility.

b. The authors mention attention maps that can be cropped and dropped, but do not provide details on how this process is done or why is it necessary. Was it explained somewhere else and I cannot find it?

c. I see that subsequently the attention cropping and dropping is explained in sub-sections 4.3.1 and 4.3.2 – why there and not before? There seems to be a lot of technical information that overlaps and makes the paper difficult to read and have an easier global understanding.

d. WSDAN seems to be the “base comparison model” the authors judge against. What is WSDAN and why is it mentioned so late in the paper and not earlier? More details on this would be beneficial.

Based on a, b, c, I think that a more “global view” of the entire methodology from the very beginning to the very end is necessary to improve understanding of what is happening – this can be done either in Figure 4 or in a different figure altogether.

4. There are other papers that have explored similar methodologies in the classification of AD that have not been cited:

C. Lian, M. Liu, L. Wang and D. Shen, "Multi-Task Weakly-Supervised Attention Network for Dementia Status Estimation With Structural MRI," in IEEE Transactions on Neural Networks and Learning Systems, vol. 33, no. 8, pp. 4056-4068, Aug. 2022, doi: 10.1109/TNNLS.2021.3055772.

M. Liu, J. Zhang, C. Lian and D. Shen, "Weakly Supervised Deep Learning for Brain Disease Prognosis Using MRI and Incomplete Clinical Scores," in IEEE Transactions on Cybernetics, vol. 50, no. 7, pp. 3381-3392, July 2020, doi: 10.1109/TCYB.2019.2904186.

Zhang et al (2021) A 3D densely connected convolution neural network with connection-wise attention mechanism for Alzheimer's disease classification, https://doi.org/10.1016/j.mri.2021.02.001

Shirkavand et al. (2021) Dementia Severity Classification under Small Sample Size and Weak Supervision in Thick Slice MRI, https://arxiv.org/abs/2103.10056

5. Small English grammar corrections throughout, please recheck – nothing serious, but still need to be taken care of.

If the above can be better argued in the paper and arguments can be brought forward, the paper can be reconsidered for publication

Reviewer 2 Report

Alzheimer's disease (AD) patients' brain lesions seem somewhat different on MRI scans than those of healthy individuals, and the categorization impact of current image recognition technology is subpar. Due to the short size of available Alzheimer's datasets,  huge neural networks. To address this issue, we offer a network model (WS-AMN) that combines weak supervision with an attention mechanism. Weakly supervised learning generates an attention map that is utilized to direct the data augmentation in step , and a channel-domain and space-domain attention module are embedded in step  of the residual network to zero in on the unique channels and spaces of pictures, respectively. The lessening of the effect of irrelevant traits is a result of the location information boosting the features that correspond to them. The results demonstrate an F1-score of 99.63 percent and an accuracy of 99.61 percent. Th authors methodology offers a state-of-the-art means of diagnosing AD with precision.

The authors need to check whether there is some overfitting in the data or not. It seems that reporting of such high accuracy might have some problems with data overfitting. As the authors stated that the data set was lower. 

Whether the data set have some behavioral scores associated? Is there any relation between the achieved accuracy and the behavioral scores?

I don't understand the purpose of table 1. Please clarify. 

Table 2: What are Val and Train? Please either write a full form or use the footnotes under the table.

The color coding in Figure is used for distinguishing the pipelines. What is the benefit of making huge blocks? The same goes for Figure 5.

In equations 18, 19, and 20, use the full form on the left-hand side of the equations. The same goes for the right-hand side. As there is no space limitation.

This heading is very strange, "Experiments and Results". Please re-think and re-write. 

Figure 10 is not clear. Please use big fonts so that they can be legible for the readers. 

Where is the discussion part, Could you please compare your results with the existing ones to validate the current results?

Author Response

See the attached document.

Round 2

Reviewer 2 Report

The authors have largely addressed this reviewer's comments, however, the added discussion section is not appropriately addressed. The authors need to include the existing studies in light of the results obtained in this work. It is extremely important to consider several published works by citing them (for an example: http://doi.org/10.2174/1567205018666210212154941) in the discussion part for the comparison of the obtained results. Please revise one more time the added discussion part for a better understanding of the readers. Also, please add the limitations of the work, to be better identified by the scientific community. 

Author Response

Response to Comments

Dear Editor,

Thank you again for your letter and the reviewer's comments on our manuscript entitled "Classification of Alzheimer's Disease Based on Weak Supervised Learning and Attention Mechanism" (ID: bransci-2007464). We carefully studied the opinions and made corrections again, hoping to get approval. Revisions are marked in yellow on the paper. The main corrections in the paper and the responses to the reviewers' comments are as follows:

Response to reviewer’s comment:

Comment:

The authors have largely addressed this reviewer's comments, however, the added discussion section is not appropriately addressed. The authors need to include the existing studies in light of the results obtained in this work. It is extremely important to consider several published works by citing them (for an example: http://doi.org/10.2174/1567205018666210212154941) in the discussion part for the comparison of the obtained results. Please revise one more time the added discussion part for a better understanding of the readers. Also, please add the limitations of the work, to be better identified by the scientific community.

Response:

According to your suggestion, I have added published papers to the discussion section. The discussion section also adds the limitations of our work, which have been marked in our paper, and the specific contents are as follows:

Compared with previous methods [19,23,38] of data augmentation and attention to extract key features, our WS-AMN is a more effective method, which can automatically focus on key feature regions for purposeful data augmentation. The experimental part demonstrates the effectiveness of our method from several aspects. Although our method achieves high accuracy in AD classification, its performance and real-world use can be further improved in the future. We only used the cross-sectional in OASIS, which is relatively single, and the two types of data are very average, without considering the impact of data differences, so we will conduct experiments with multiple data sets in the future. Our network currently only focuses on the classification of AD and NC. In the future, mild cognitive impairment (MCI) should also be taken into account. MCI is an intermediate stage between NC and AD, with a high probability of turning to AD. Therefore, accurate detection of MCI can further effectively prevent AD. Our network can now achieve a classification accuracy of  for AD. In the future, pipelines should be developed for average users. By monitoring brain changes, we can predict whether they will develop AD in the upcoming 10 years.

We tried our best to improve the manuscript and made some changes in the manuscript. We appreciate for Editors and Reviewers’ warm work earnestly, and hope that the correction will meet with approval. Thank you very much for your comments and suggestions. We would like to thank the reviewers again for taking the time to review our manuscript.